# p38γ MAPK Inflammatory and Metabolic Signaling in Physiology and Disease

**DOI:** 10.3390/cells12131674

**Published:** 2023-06-21

**Authors:** Xiao-Mei Qi, Guan Chen

**Affiliations:** 1Department of Pharmacology and Toxicology, Medical College of Wisconsin, Milwaukee, WI 53226, USA; 2Research Service, Clement J. Zablocki Veterans Affairs Medical Center, Milwaukee, WI 53295, USA

**Keywords:** p38γ, signal transduction, cancer

## Abstract

p38γ MAPK (also called ERK6 or SAPK3) is a family member of stress-activated MAPKs and has common and specific roles as compared to other p38 proteins in signal transduction. Recent studies showed that, in addition to inflammation, p38γ metabolic signaling is involved in physiological exercise and in pathogenesis of cancer, diabetes, and Alzheimer’s disease, indicating its potential as a therapeutic target. p38γphosphorylates at least 19 substrates through which p38γ activity is further modified to regulate life-important cellular processes such as proliferation, differentiation, cell death, and transformation, thereby impacting biological outcomes of p38γ-driven pathogenesis. P38γ signaling is characterized by its unique reciprocal regulation with its specific phosphatase PTPH1 and by its direct binding to promoter DNAs, leading to transcriptional activation of targets including cancer-like stem cell drivers. This paper will review recent findings about p38γ inflammation and metabolic signaling in physiology and diseases. Moreover, we will discuss the progress in the development of p38γ-specific pharmacological inhibitors for therapeutic intervention in disease prevention and treatment by targeting the p38γ signaling network.

## 1. Introduction

p38γ mitogen-activated protein kinase (MAPK) is an isoform of p38 family proteins (p38α, β, γ, and δ) that are encoded by four different genes in different chromosomes. p38α (Gene name: *MAPK14*) and p38β (*MAPK11*) are expressed ubiquitously, whereas p38γ (*MAPK12*) and p38δ (*MAPK13*) are expressed in certain tissues (for example, p38γ in skeletal muscle; and p38δ in the salivary, pituitary, and adrenal glands) [1]. All p38s contain a conserved Thr–Gly–Tyr (TGY) dual-phosphorylation motif within the kinase activation loop, and both Thr and Tyr phosphorylation are necessary to fully activate the kinase. p38 MAPKs are phosphorylated and activated by MAPK kinase kinase 3 (MKK3) and/or MKK6, which in turn phosphorylate a substrate containing an ST/P motif [1]. While p38α and p38β can directly phosphorylate more than 100 substrates [2], p38γ and p38δ have specific and non-overlapping substrates [3,4]. In response to stress and inflammatory stimuli, p38α is most frequently activated, while other p38 family members are phosphorylated by a mechanism depending on cell type and/or stimuli. Although these isoform-specific activations are important for different cellular outcomes, the mechanisms involved are largely unknown [2,5,6]. Isoform-specific and tissue-dependent effects of p38 MAPKs in inflammation and inflammation-associated oncogenesis have been recently reviewed [7]. This manuscript will focus on recent discoveries regarding p38γ inflammatory and metabolic signaling in physiology and diseases.

p38γ was cloned in 1996 [8,9,10] and is about 60% homologous to p38α. A three-dimensional (3D) search of active CDK1 and CDK2 revealed a higher degree of structural similarity with p38γ than with other stress kinases [11]. P38γ is unique among MAPKs due to its C-terminal sequence, called the PDZ motif, which can bind to PDZ domain proteins, which may be the determinant for its distinct activity as compared to its close family member p38δ. This allows p38γ to form PDZ-dependent protein complexes essential for its specific activities. Studies about the role of PDZ binding in p38γ oncogenic signaling and biological activities have been recently reviewed [12]. Recent studies showed that p38γ is involved in the pathogenesis of several important diseases including diabetes [13], cancer [11], and Alzheimer disease [14] through phosphorylating various substrates. We will discuss these findings on p38γ as well as potential implications and possible mechanisms [4,12]. Targeting the p38γ signaling network may, therefore, be a novel approach in disease prevention and treatment.

## 2. p38γ Signaling in Inflammation

p38γ protein is initially reported as undetectable in inflammatory cell lineages as compared to its family proteins [15]. In patient tissues of rheumatoid arthritis, immunological studies showed that p38γ and p38α are up-regulated as compared with other p38 isoforms by using isoform-specific antibodies [16]. Transiently transfected p38γ in KB or 293 cells is strongly activated by a pulse exposure to stress stimuli and cytokines [17]. In a separate study, endogenous p38γ in chondrocytes was phosphorylated in response to IL-1β similarly to p38α, ERK1, and JNK2, and ectopic expression of a constitutively active p38γ alone decreased MMP-13 (collagenase-3) expression [18]. While IL-1β and TNFα require both p38γ and other p38 family members to stimulate disintegrin and metalloproteinase with thrombospondin motifs 4 (ADAMTS4 gene) expression in nucleus pulposus (NP) cells [19], the kinetics of p38γ activation are different than that of p38α in 293T cells after IL-1β, indicating their distinct roles in cytokine signaling [20]. p38γ- or p38δ-deficient mice (^−/−^), however, have reduced arthritis and their double knockout (p38γ/δ^−/−^) mice further have lower levels of pathogenic anti-collagen antibodies and IL-1β and TNFα cytokines than wild-type mice in the collagen-induced arthritis (CIA) model [21]. Thus, although p38γ may not be intrinsically detected in some cells, it can be activated similarly to other p38 proteins in a cell and/or tissue specific manner in response to inflammatory signals and it can regulate cytokine signaling similarly to p38δ and cooperate with p38δ in the inflammatory response.

This conclusion is further supported by genetic studies. Mice with whole-body knockout of p38γ gene appear normal [22]. In response to lipopolysaccharide (LPS), however, mice with p38γ and/or p38δ deletion (p38γ^−/−^ and/or p38δ^−/−^) showed an impaired innate immune response and decreased ERK activation, attenuated production of TNFα, IL-1β, and IL-10 in macrophages, and increased resistance to septic shock [23]. Myeloid-specific knockout of p38γ and p38δ decreases the LPS-induced serum concentration of TNFα and IL-6 [24], and inhibits the expression of tumor progression locus 2 (TPL2) in multiple organs [25]. In a study of T cell development, p38γ deficiency favors thymocyte positive selection from CD4^+^CD8^+^ double-positive to CD4^+^ or CD8^+^ single-positive cells [26]. While p38γ/δ deletion does not affect the B cell differentiation program in bone marrow, it reduced numbers of peripheral B cells and altered marginal zone B cell differentiation in the spleen [27]. In addition, the expression of co-stimulatory proteins and activation markers in p38γ/δ-deficient B cells are diminished in response to B cell receptor (BCR) and CD40 stimulation. Further, p38γ and p38δ were necessary for B cell proliferation induced by BCR and CD40, but not by TLR4 signaling, and p38γ/δ-null mice produced significantly lower antibody responses to T-dependent antigens [27]. Bone marrow-derived macrophages (BMDM) from p38γ and p38δ double knockout mice produced much less cytokines and chemokines, including TNF, IL-6, IL-β, IL-10, MIP-2, and CCL2, in response to *C. albicans* infection, and p38γ/p38δ deficiency decreased the inflammatory response against *candida albicans* infection and increased macrophage and neutrophil antifungal activity [28]. These results together indicate that p38γ may be involved in T and/or B cell development, cytokine/chemokine expression, antibody production, and outcomes of chronic inflammation.

Recent studies showed that p38γ is involved in inflammation-associated cancer. In an Azoxymethane (AOM)/Dextran-Sulphate-Sodium (DSS) AOM/DSS model, DSS-induced RNA expression of IL-1β and IL-6 in colon tissues was decreased in whole-body p38γ knockout (p38γ^−/−^) mice, leading to decreased tumor formation [29]. In the same study, double-knockout of p38γ and p38δ further decreased tumorigenesis as compared to p38γ^−/−^ alone [29], indicating that p38γ and p38δ may cooperate to promote colitis-associated cancer. In a 7,12-dimethylbenz[*a*]anthracene/12-Otetradecanoylphorbol-13-acetate (DMBA/TPA) mouse skin cancer model, there is a decrease in cytokine levels in p38γ and p38δ double knockout mice (p38γ/p38δ^−/−^), with decreased tumor formation, but the contribution by p38γ alone, however, remains unknown [30]. Studies from our lab showed that DSS treatment stimulates p38γ, but not p38α; and phosphorylation in colon tissues and deletion of p38γ from intestinal epithelial cells (p38γ KO) reduces colitis and colitis-associated cancer in an AOM/DSS mouse model [31], indicating a required role of epithelial p38γ in inflammation-driven colon cancer. Importantly, this study showed a decrease in cytokine (TNF, IL-1β, and IL-6) induction and Wnt signaling in p38γ KO mice and the application of p38γ inhibitor pirfenidone (PFD) only decreases tumorigenesis and reduces cytokine expression in wild-type, but not p38γ KO, mice, indicating its dependence on epithelial p38γ to suppress inflammation-induced colon cancer [31]. Moreover, liver-specific knockout of p38γ blocks carcinogen-induced liver cancer [11]. Conditional deletion of p38γ from pancreas epithelial cells also inhibits cerulein-induced cytokine expression and pancreatitis, and suppresses pancreatic tumorigenesis in KPC mice [32]. These genetic studies together demonstrate that systemic and epithelial p38γ is required for the inflammatory response and for tumorigenesis of colon, liver, and pancreas in the gastrointestinal tract.

## 3. p38γ Signaling in Metabolism

Four p38 isoforms (α, β, γ, and δ) exist in both the human and mouse genomes [33] but only p38γ is predominantly expressed in skeletal muscle [8]. One early study showed that p38γ overexpression promotes, whereas expression of its dominant negative form inhibits, myogenic differentiation in C2C12 muscle cells [9]. In a separate study using C2C12 myogenic cells as a model, p38α, β, and γ were found to be expressed with distinct patterns during differentiation, and knockdown of any of them inhibits myogenic differentiation with distinct effects on gene expression [34]. Studies with muscle-specific deletion of all four p38 genes, however, showed that only p38γ, but not p38α, p38β, or p38δ, is required for endurance exercise-induced metabolic adaption [35], indicating a required role of p38γ in physiological exercise. Moreover, p38γ and/or p38δ regulate the mammalian target of rapamycin (mTOR) pathway, and control protein synthesis and heart growth [36]. Mice with myeloid-specific deletion of p38γ and p38δ are protected against steatohepatitis and fibrosis, and are resistant to diet-induced fatty liver, hepatic triglyceride accumulation and glucose intolerance [37], indicating a role of hematopoietic p38γ and/or p38δ in regulating liver metabolism. Of great interest, p38γ/p38δ-deficient neutrophils are defective in migration to the damaged liver in association with a decreased induction of cytokines IL-6 and TNF by a methionine–choline-deficient (MCD) diet [37]. Moreover, p38γ negatively regulates asymmetric satellite stem cell division and is required for symmetric self-renewal, and satellite cell-specific genetic deletion of p38γ impairs muscle regeneration [38]. Cardiac-specific p38γ/p38δ overexpression leads to glucose intolerance and insulin resistance, whereas early postnatal cardiac-specific p38γ and p38δ deletion increases cardiac glycogen storage and affects whole-body metabolism [39]. In addition, p38γ knockdown suppressed lipid accumulation by inhibiting the JAK-STAT pathway, suggesting that targeting p38γ may contribute to the suppression of lipid accumulation in fatty liver disease [40]. The results show that p38γ plays an important role in regulation of glucose and lipid metabolism in the heart, muscles, and liver.

Glucose-regulatory effects of p38γ may also be involved in the pathogenesis of diabetes and cancer. In a study of metabolic interactions between the gut microbiota and diet, imidazole propionate, as a microbially produced metabolite, was found to impair glucose intolerance and insulin signaling at the level of insulin receptor substrate through the activation of p38γ [13]. Increased aerobic glycolysis (Warburg effect) is a hallmark of cancer. The KRAS oncogene stimulates glycolysis in order to transform cells and to induce tumorigenesis, but the mechanisms are largely unknown [41]. We recently showed that the KRAS oncogene directly stimulates p38γ expression and phosphorylation in pancreatic epithelial cells, in which p38γ interacts with several metabolic proteins including phosphofructokinase-2/fructose-2,6-bisphosphatase (PFKFB3) [32]. Further analysis showed that p38γ is required for KRAS-induced glucose uptake and lactate secretion in cancer cells through PFKFB3 and glucose transporter 2 (Glut2) [32], whereas an earlier study showed that p38γ promotes glucose uptake by increasing Glut1 expression in fibroblasts [42]. These results indicate that p38γ can promote glucose uptake in different tissues via different glucose transporters. p38γ also activates metabolic pathways in 231 human breast cancer cells as demonstrated by both knockdown and overexpression [43]. These studies together demonstrate a critical role of p38γ in diabetes and cancer by regulation of glucose-associated metabolic signaling.

## 4. p38γ Signaling to Its Substrates in Physiology and Diseases

p38γ is a serine/threonine kinase and both its expression and phosphorylation are important for its inflammatory, metabolic, and oncogenic signaling (Figure 1). In response to the KRAS oncogene, p38γ is induced in rat intestinal epithelial cells (IEC-6) but not in mouse NIH3T3 fibroblasts [44,45]. In pancreatic epithelial HPNE cells, however, KRAS transformation increases both p38γ protein expression and phosphorylation. These results suggest that the KRAS oncogene activates p38γ through increased expression and elevated phosphorylation by a mechanism depending on cell types and tissue origin. Because p38γ phosphorylation is decreased by transient KRAS co-transfection in IEC-6 cells [44] and p38γ is the only MAPK that contains a PDZ-domain binding motif at its C terminus [46], we sought to search for its specific phosphatase using a PDZ-based two-hybrid screening. We found that wild-type p38γ, but not its PDZ-deleted mutant, interacts with a PDZ-domain containing protein tyrosine phosphatase H1 (PTPH1), which decreases p38γ phosphorylation in vitro and in vivo [47]. Through PDZ binding, p38γ was previously shown to phosphorylate SAP97/S122 [22], which is implicated in ethanol-activated and p38γ-dependent stimulation of the cancer-like stem cell (CSC) population and breast cancer growth [48]. Of interest, PDZ binding is required for p38γ [44,47,49] and PTPH1 [50,51,52] oncogenic activity [53], as expression of their PDZ-binding-deficient mutants by stable transfection or application of peptide to disrupt the endogenous p38γ/PTPH1 interaction inhibits cancer cell growth [47]. The same PDZ binding also enables p38γ phosphorylation of PTPH1 at S459 in vitro and in vivo, whereas p38α lacks this activity, which is required for PTPH1 phosphatase and oncogenic activity [50]. These results indicate a critical role of the PDZ complex in oncogenesis, which requires phosphorylation/de-phosphorylation of p38γ and PTPH1. Consistent with this speculation, after tetracycline-inducible KRAS expression (tet-on system), upregulated p38γ is dominant and persistent, whereas the resultant p-PTPH1/S459 is delayed and transient [54]. This leads to increased EGFR protein expression (due to p38γ) and enhanced EGFR de-phosphorylation (due to PTPH1), resulting in a phenotype of increased levels of un-phosphorylated EGFR proteins in KRAS-mutant colon cancer cells [54]. Because KRAS induces the protein expression of p38γ and PTPH1, this reciprocal p38γ dephosphorylation and PTPH1 phosphorylation may be not only important for transformation but for certain phenotypes of KRAS-dependent tumors such as those resistant to EGFR inhibitors [12].

In cancer cells, p38γ phosphorylates additional substrates. In breast cancer cells p38γ phosphorylates estrogen receptor α (ER) at S118 and inhibits its proteasome-dependent degradation [55]. This phosphorylation event at ER/S118 enables p38γ to cooperate with c-Jun in binding the cyclin D1 promoter, leading to increased cyclin D1 expression and decreased breast cancer sensitivity to anti-estrogen Tamoxifen (TAM) [55]. Additionally, p38γ phosphorylates DNA Topo IIα/S1542 in vitro and in vivo, thereby increasing Topo II stability and catalytic activity as well as breast cancer cell sensitivity to Topo II inhibitors [56]. In addition, p38γ-phosphorylated protein phosphatase PTPH1 can dephosphorylate ER at Y539 and increase nuclear ER accumulation and enhance breast cancer sensitivity to anti-estrogens [57]. Moreover, p38γ can cooperate with c-Jun to stimulate epidermal growth factor receptor (EGFR) transcription [54]. PTPH1 can also dephosphorylate EGFR in breast cancer cells and disrupt EGFR interaction with ER in the cytoplasmic membrane and thereby sensitize breast cancer cells to EGFR inhibitor lapatinib [52]. Thus, p38γ can directly phosphorylate substrates in cancer cells to impact cellular outcomes and indirectly regulate cancer cell growth via phosphorylating its substrate PTPH1 that is required for de-phosphorylation of proliferative proteins (Figure 1).

Several studies showed that p38γ can phosphorylate several oncogenic proteins in cancer cells and/or tissues, and that is implicated in tumor-growth in mice. Proteomic screening identified that p38γ specifically interacts with heat shock protein 90 α (HSP90α) in KRAS-mutated but not KRAS wild-type colon cancer cells, and phosphorylates HSP90α at S595, thus preventing mutant KRAS protein from degradation and conferring sensitivity of KRAS mutant colon cancer cells to HSP90 inhibitor 17-alllylaminogeldanamycin (17-AAG) [58]. Application of the p38γ inhibitor pirfenidone (PFD) blocked p38γ-induced HSP90/S595 phosphorylation, promoted mutant KRAS degradation, and inhibited colon cancer xenograft growth in nude mice [58]. β-catenin is a key member of the Wnt pathway and studies by Yin et al. found that intestinal epithelial cell (IEC)-specific knockout of p38γ suppresses the β-catenin/Wnt pathway [31]. Mechanistic analysis further showed that p38γ binds and phosphorylates β-catenin at S605, which is important for β-catenin stability, and that PFD requires epithelial p38γ to inhibit phosphorylation of several oncogenic substrates including PTPH1, Hsp90, and β-catenin to inhibit colon tumorigenesis [31]. These results indicate that p38γ may cooperate with several substrates to promote colon cancer development and growth through phosphorylation (Figure 1). p38γ-induced SAP97/S122 may be also involved in mammary tumor development and growth in vitro [22,48]. In pancreatic cancer, p38γ binds phosphofructokinase-2/fructose-2,6-bisphosphatase 3 (PFKFB3) in KRAS-transformed cells and phosphorylates PFKFB3 at S467 in human pancreatic cancer cells and in mouse KPC tumors [32]. Moreover, p-PFKFB3 protein levels were only completely depleted in p38γ-expressing KPC tumor cells by a combination of the p38γ inhibitor PFD with the PFKFB3 inhibitor PFK15, and expression of the phosphorylation mutant PFKFB3/S467A in KPC tumors decreased the tumor growth in nude mice, as compared to the tumor expressing wild-type PFKFB3. These results demonstrate that epithelial p38γ may be a systemic target for therapeutic intervention through phosphorylating its substrates.

Studies further showed that p38γ can phosphorylate tumor suppressors and thereby regulate tumorigenesis. p38γ phosphorylates the retinoblastoma (Rb) at S807/S811 through which p38γ compensates for the loss of CDK1 or CDK2, and both p38γ conditional knockout (KO) and the application of PFD significantly prevent carcinogen-induced liver tumor tumorigenesis [11]. Moreover, p38γ, as well as p38α, can phosphorylate p53/S33 and mediate RAS oncogene-induced senescence, but only p38γ, but not p38α, is essential for oncogenic Ras-induced transcriptional activity of p53 [59]. Whether p38γ phosphorylating p53 impacts transformation and tumor growth in murine models has not been demonstrated.

p38γ also cooperates with its substrates to impact other pathological processes. In a study of neurological disorders, p38γ was found to phosphorylate tau protein at T181, which together with extracellular amyloid-β (AB), orchestrates neuronal dysfunction in Alzheimer’s disease, whereas a site-specific tau phosphorylation disrupts the PSD-95/tau/Fyn interaction and inhibits Aβ toxicity [14]. In a study of muscle stem cell commitment, p38γ phosphorylates coactivator-associated arginine methyltransferase 1 Carm1 at S572 to prevent its nuclear translocation, which functions to oppose the activity of p38α and stimulates self-renewal rather than differentiation [38]. In a study of heart hypertrophy, however, p38γ and p38δ phosphorylate mTOR complex 1 (mTORC1) and mTOR complex 2 (mTORC2) inhibitor DEPTOR at S145, S244, and S265, promoting its downregulation [36]. Furthermore, p38γ phosphorylates MyoD at S199/S200, thereby enhancing MyoD occupancy on the Myogenin promoter to form a repressive transcriptional complex [60].

Further, p38γ can phosphorylate insulin receptor substrate p62 at T269/S272, which may contribute to type 2 diabetes [13]. During heart development, p38γ and p38δ contribute to the cardiac metabolic switch through inhibitory phosphorylation of glycogen synthase 1 (GYS1) at S723, S727, and T278, leading to glycogen metabolism inactivation [39]. Calpastation is also phosphorylated by p38γ at T216/S219, resulting in its reduced activity to inhibit the protease calpain in ventricular remodeling [61]. SAP90 is phosphorylated by p38γ and ERK2 at T287 and S290, which may be important for their co-localization in neurons [62]. In addition, p38γ phosphorylates α1-syntrophin at S193/S201 though PDZ binding to regulate its localization and substrate specificity [63]. While several p38 MAPKs can regulate heat shock factor 1 (HSF1), p38γ is the principal isoform responsible for its phosphorylation at S326 in cells, which affects the extent and duration of the heat shock response [64]. These results together indicate that p38γ may cooperate with its specific and common substrates and other p38s and/or MAPKs to form a signaling network through phosphorylation to regulate oncogenesis, diabetes, heart hypertrophy, and neurological disorders (Figure 1).

## 5. p38γ Specific Inhibitors

There have been great efforts to develop p38γ-specific pharmacological inhibitors over several decades. The first p38γ inhibitor pirfenidone (PFD) was developed by interMune, with its patent recently sold to Genentech. This compound showed a selective activity to inhibit p38γ over p38α or p38β in vitro, with the goal to develop an anti-fibrotic drug for lung fibrosis [65]. Due to its good clinical efficacy and low toxicity, PFD has been FDA-approved for the treatment of patients with lung fibrosis [66], although whether p38γ is the target for its anti-fibrotic actions is still unknown. In cancer growth-inhibitory studies, a high concentration of PFD is needed in cell culture and in animals to achieve the desired effect (Table 1). PFD is used daily for two months at 500 mg/kg to inhibit tumorigenesis in KPC mice [32]. A pan-p38 inhibitor BIRB796 [67,68,69] was developed later, which has more potent activity in inhibiting p38α in vivo, albeit it is more potent at inhibiting p38γ activity in vitro. In a mouse xenograft study, a report showed that BIRB796 (10 mg/kg) suppresses xenograft growth in nude mice [68]. Through a high-throughput kinase inhibitor screen, PIK75 was identified and it has strong p38γ inhibitory activity with an IC_50_ of about 0.34 to 0.42 μM. Although it shows a strong p38γ inhibitory activity as compared with other p38 MAPK proteins (>0.1 μM) [70], PIK75 also significantly inhibits PI3K activity (IC50: 0.075 μM). PIK75 can suppress the xenograft growth in mice at a low dose (2 mg/kg) [70]. The same group recently further synthesized a new compound CSH71 targeting the non-ATP-binding pocket of p38γ by binding the lipid-binding-domain (LBD). CSH71 has strong cytotoxicity in Hu78 cells (cutaneous T-lymphocyte cell line) with an IC_50_ of about 2 μM, and its in vivo anti-tumor activity as well as its potential effect on p38δ remains untested [71]. Another recent study by using phenotype-based screens with conformation-specific inhibitors found that AD80 is a new inhibitor for p38γ and p38δ in hepatocellular carcinoma (HCC) [72]. AD80 inhibits the growth of a panel of HCC cell lines at μM range and suppresses the xenograft growth at 20 mg/kg with stronger inhibitory activity on p38γ and p38δ than p38α and p38β, by an in vitro kinase assay [72]. Thus, while progress has been made, there is still a great need to develop specific p38γ pharmacological inhibitors for therapeutic exploration.

## 6. Perspectives

p38γ is a unique MAPK with a C-terminal PDZ-binding motif, through which it interacts with a variety of proteins including its substrates, phosphatase, and transcription factors. This structure may be the key factor for p38γ signaling to its substrates in the regulation of inflammation, metabolism, and pathological processes of cancer, diabetes, heart hypertrophy, and neurological disorders (Figure 1). Some of p38γ’s biological activities are specific and even opposite to p38α [6,7,73,74,75]. In addition to stress and inflammation-induced phosphorylation, p38γ can be up-regulated by the RAS oncogene and suppressed by the nuclear receptor ER in epithelial cells [32,44,76]. Moreover, p38γ can phosphorylate specific substrates (Figure 1) through which p38γ signaling can be further amplified or modified to fine-tune biological outcomes. Genetic and pharmacological studies showed that p38γ may be a therapeutic target in cancer. Targeting the p38γ signaling network would therefore have important implications in disease prevention and control.

p38γ oncogenic activity has been extensively studied for the past 20 years (see review [77]. A recent hallmark paper found that p38γ is required for liver tumorigenesis and phosphorylates Rb protein [11], which thus links the MAPK pathway activity with cell cycle progression in oncogenesis. Because p38γ also phosphorylates tumor suppressor p53 and oncogenic proteins β-catenin, DNA Topo IIα, PFKFB3, and ERα, p38γ may act through activating phosphorylation of oncogenic proteins and inactivating phosphorylation of tumor suppressors to integrate their growth regulatory activities. Disruption of p38γ’s signaling to its substrates by specific p38γ inhibitors and gene knockout or silencing will further improve our understanding how p38γ and its substrates cooperate to promote oncogenesis, which may be fundamental in developing effective targeted cancer therapies.

Mechanistically, p38γ activity is linked to two important signaling processes that may determine cellular outcome. One is the reciprocal kinase–phosphatase crosstalk between p38γ and PTPH1, in which p38γ phosphorylates PTPH1 and PTPH1 dephosphorylates p38γ, which may cooperate to promote KRAS oncogenesis [47,50]. However, this crosstalk may also restrict p38γ activity to phosphorylate and/or to enhance PTPH1 dephosphorylating oncogenic substrates such as EGFR to confer the phenotype of KRAS-mutant colon cancer resistant to EGFR inhibitors [54]. Another is the intrinsic activity of p38γ to bind gene promoters, thereby stimulating gene transcription that may be a critical factor to activate the stem cell program in several systems [38,78,79,80]. Studies have shown that, through interaction with c-Jun, p38γ can bind to the AP-1 site of MMP9 [81], EGFR [54], cyclin D1 [55], and Nanog [78] to stimulate their transcription and to increase cell growth or motility. Because phosphorylated p38γ is mostly in the nucleus [74], nuclear p38γ is likely more oncogenic. This assumption is also supported by the fact that p38γ is decreased in ER^+^ breast cancer [55,76] but acts as an oncogene and therapeutic target in triple-negative breast cancer [78].

Although mice with whole-body knockout of p38γ (p38γ^−/−^) are normal [22], these animals are resistant to chemical-induced colon and skin cancer [29,30]. Moreover, p38γ^−/−^ mice have impaired innate immune responses to lipopolysaccharide (LPS) and a Toll-like receptor 4 (TLR4) ligand, and had reduced TNFα, IL-1β, and IL-10 production in macrophages [23]. Studies with a whole-body and conditional knockout further showed that p38γ and p38δ are both important for B cell development and antibody production to T-antigens [27], while myeloid-specific p38γ deletion increases antifungal activity [28]. Deletion of the p38γ gene from the epithelial cells of GI tissues showed that p38γ is required for cytokine expression, and colon, liver, and pancreatic tumorigenesis [11,31,32]. These results together indicate that p38γ plays a key role in immune cell development and GI tumorigenesis. Future studies are warranted to investigate how p38γ activity in immune and epithelial cell compartments cross-talk to impact tumorigenesis through its inflammatory and metabolic signaling.

## Figures and Tables

**Figure 1 cells-12-01674-f001:**
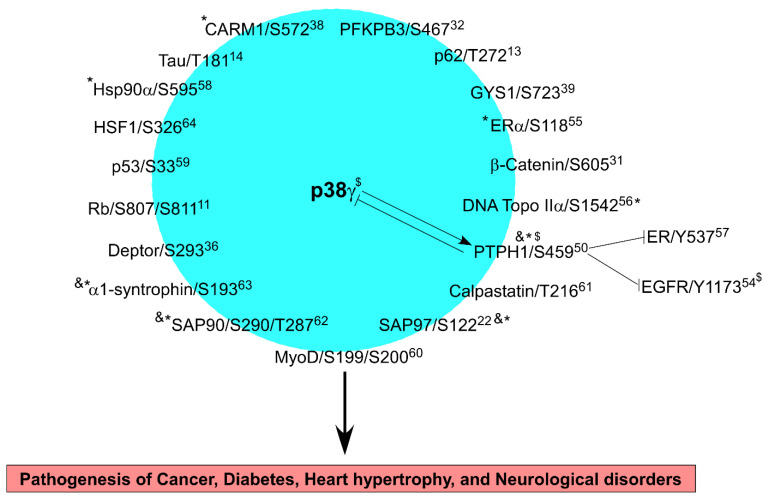
p38γ signaling network in diseases. The number after each protein indicates the reference number in which the phosphorylation was demonstrated. The reciprocal effects of p38γ and PTPH1 were illustrated with PTPH1 substrates (EGFR and ER) also listed. p38γ may cooperate with one or more substrates to impact pathogenesis of a disease (highlighted inside the box below) and one phosphorylation event may be involved in several diseases. * indicates a specific phosphorylation by p38γ and not by its family member p38α, whereas & shows a PDZ-dependent reaction. $ indicates that EGFR depends on p38γ PDZ motif and phosphorylation to form a complex with both p38γ and PTPH1.

**Table 1 cells-12-01674-t001:** Experimental p38γ pharmacological inhibitors.

Inhibitor Name	Selectivity	Concentration In Vitro; Dose In Vivo	Refs.
Pirfenidone (PFD)	p38γ > α > β	100–400 μg/mL (0.54–2.16 mM); 500 mg/kg	[32,65,66]
BIRB796	p38α > β > γ > δ	0.1–10 μM; 10 mg/kg	[67,68,69]
PIK75	PI3K and p38γ	0.01–1 μM; 2–10 mg/kg	[70]
AD80	p38γ and p38δ	0.01–10 μM; 20 mg/kg	[72]
CSH71	p38γ	0.01–1 μM	[71]

## Data Availability

Available from the PI upon reasonable request.

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
