# Peer review of "p38γ MAPK Inflammatory and Metabolic Signaling in Physiology and Disease"

_cells, 2023, doi:10.3390/cells12131674_

Round 1

Reviewer 1 Report

This is a very nice review but feel some parts need some modifications,  I hope the comments from my review will help.

Very reasonable,  some corrections needed

Author Response

Outstanding review and thank you for your time and effort!

Reviewer 2 Report

This paper reviews published work on the p38gamma MAPK, mostly related to inflammation and metabolic regulation. It also discusses a number of reported substrates, as well as several chemical inhibitors.

1. The manuscript would benefit from a bit more elaboration of the ideas presented, as it often reads as a list of results from various publications without drawing any clear conclusion or trying to explain the discrepancies between studies.

2. It would be interesting to add at least one Table or Figure showing the different p38g substrates that are known, together with the phosphorylation sites and the functions reported in each case.

3. One peculiarity of p38g is that it contains a C-terminal PDZ-binding motif, and it would be very useful to indicate which substrates or regulators interact with p38g through this domain. Likewise, it should be stated which p38g regulated functions have been shown to require the PDZ-binding motif. This information could be added to the above Table or explained in a dedicated section of the text.

4. The review would be more valuable if it addressed the functional interplay between p38g and other p38 MAPKs. For example, by cataloguing the biological functions in which p38g is likely to play redundant roles with p38d, and those in which the evidence suggests that p38g and p38d play different roles. In the same line, the biological functions in which p38g may perform overlapping roles with p38a, and those in which p38g and p38a could actually play opposite roles could be explained. This information could be discussed in a dedicated section of the text, or schematically mentioned in a Table or Figure.

5. The text refers to a Figure 1 that is not included in the manuscript.

6. Table 1 should indicate the in vitro concentration of PFD in microM to compare with the other inhibitors.

7. The work by the authors tend to be described more extensively, even when referring to rather old publications, than the work of other groups. This should be avoided.

8. Lines 325/326 state: “.. which, for the first time, links MAPK signaling pathway to cell-cycle progression”. This is not correct as a number of previous papers have linked signaling by MAPKs, including p38 MAPKs, to cell-cycle progression.

9. The text contains several typos and grammatical mistakes and should be carefully proof-read.

The text contains several typos and grammatical mistakes, and should be carefully proof-read.

Author Response

Thank you for the good suggestion about writing more p38g C-terminal PDZ motif.

Reviewer 3 Report

     It is suggested to write the manuscript in a more concise and informative way. For example, the sentence this study showed a decrease in inflammation-induced colon cancer (lines 102-106) was way too long.

     p38γ and p38δ 70% identical to each other, and the authors also mentioned lots of studies regarding the roles of p38γand p38δ. Please emphasize the rationale that p38γshould be focused on in this review and whether is there any difference between p38γ and p38δ in terms of their function.

     Figure 1 is lacking. Also, please check the reference format (e.g. {Shin, 2005 #2642} in line 170).

     It is suggested to divide the content into various subdivisions with proper subtitles. Besides, it will be helpful if the authors can provide a table with the downstream regulators of p38γ, disease types, or their main functions. Otherwise, it is hard for the readers to extract the information they need.

     It is pretty odd to present a paragraph on the p38γ inhibitors abruptly.

     Please double-check the grammar. For instance, in line 33 p38β can directly phosphorylates more than 100 substrates.

Author Response

Thank you for your suggestion of more writing about p38g and p38d comparison.

Reviewer 4 Report

The manuscript" p38 MAPK inflammatory and metabolic signaling in physiol- 2 ogy and disease" need some improvements. 

Need to carefully read line by line, and correct spellings and several English grammar mistakes. 

The introduction section is very short, and can be improve by adding the more informatics studies. I refer some good published studies to read, and add text in introduction, and cite it.

such as : https://doi.org/10.3892/mmr.2023.12940

https://doi.org/10.3390/ijms231911746

doi: 10.3389/fgene.2022.1008502

https://doi.org/10.3389/fonc.2022.899009

doi: 10.4081/ejh.2022.3415

Add relevant figures to explain the pathways. only writing the text is not enough. 

Please add the chemical structures of inhibitors mentioned in the manuscript. (use as Chemdraw or Chemsketch software).

Add table of inhibitors extracted from natural sources too. It will improve the study. 

Overall paper is not bad, but need improvements. 

Moderate English grammar and spelling mistakes are there in the Manuscript. 

Author Response

Thank you about your suggestion of including more background information and structures of p38g inhibitors.

Round 2

Reviewer 2 Report

I recommend publication

OK

Reviewer 3 Report

N/A

N/A

Reviewer 4 Report

The authors have modified the paper and incorporated the answer to previous comments, and now it's very much improved. 

The paper can be accepted now.